# Gut Microbiota and Short Chain Fatty Acids: Implications in Glucose Homeostasis

**DOI:** 10.3390/ijms23031105

**Published:** 2022-01-20

**Authors:** Piero Portincasa, Leonilde Bonfrate, Mirco Vacca, Maria De Angelis, Ilaria Farella, Elisa Lanza, Mohamad Khalil, David Q.-H. Wang, Markus Sperandio, Agostino Di Ciaula

**Affiliations:** 1Clinica Medica “A. Murri”, Department of Biomedical Sciences & Human Oncology, University of Bari “Aldo Moro”, 70124 Bari, Italy; ilaria.farella@uniba.it (I.F.); elisa.lanza@uniba.it (E.L.); mohamad.khalil@uniba.it (M.K.); agostinodiciaula@tiscali.it (A.D.C.); 2Department of Soil, Plant and Food Sciences, University of Bari Aldo Moro, Via Amendola 165/a, 70126 Bari, Italy; mirco.vacca@uniba.it (M.V.); maria.deangelis@uniba.it (M.D.A.); 3Department of Medicine and Genetics, Division of Gastroenterology and Liver Diseases, Marion Bessin Liver Research Center, Einstein-Mount Sinai Diabetes Research Center, Albert Einstein College of Medicine, Bronx, NY 10461, USA; david.wang@einsteinmed.edu; 4Biomedical Center (BMC), Institute for Cardiovascular Physiology and Pathophysiology, Walter Brendel Center for Experimental Medicine (WBex), Faculty of Medicine, Ludwig-Maximilians-Universität Munich, 82152 Planegg-Martinsried, Germany; markus.sperandio@lmu.de

**Keywords:** intestine, bacteria, metabolome, fiber, diet, glucose homeostasis

## Abstract

Gut microbiota encompasses a wide variety of commensal microorganisms consisting of trillions of bacteria, fungi, and viruses. This microbial population coexists in symbiosis with the host, and related metabolites have profound effects on human health. In this respect, gut microbiota plays a pivotal role in the regulation of metabolic, endocrine, and immune functions. Bacterial metabolites include the short chain fatty acids (SCFAs) acetate (C2), propionate (C3), and butyrate (C4), which are the most abundant SCFAs in the human body and the most abundant anions in the colon. SCFAs are made from fermentation of dietary fiber and resistant starch in the gut. They modulate several metabolic pathways and are involved in obesity, insulin resistance, and type 2 diabetes. Thus, diet might influence gut microbiota composition and activity, SCFAs production, and metabolic effects. In this narrative review, we discuss the relevant research focusing on the relationship between gut microbiota, SCFAs, and glucose metabolism.

## 1. Introduction

The gastrointestinal tract hosts a large interface between the external environment and the human body. The gut surface is the largest of human body and extends for roughly 200–300 m^2^ [1]. At this level, the gut microbiota represents a complex polymicrobial ecology and a dynamic ecosystem. More than 100 trillion bacteria in the human gut operate symbiotically with the host and with diverse external stimuli. Products of gut microbial degradation of nutrients are bioactive metabolites that bind target receptors, activate signaling cascades, and modulate several metabolic pathways with local and systemic effects [2]. 

Short chain fatty acids (SCFAs) are among the important class of gut microbiota bio-products, produced mainly from fermentation of non-digestible carbohydrates, including dietary fiber, that become available to the gut microbiota [3]. Although fiber fermentation plays an essential role in the modulation of gut microbiota physiology and composition, SCFAs also impact on host health at cellular, tissue, and organs levels by mechanisms related to gut barrier function, glucose homeostasis, immunomodulation, and obesity [4]. Thus, the gut microbiota plays a critical role in maintaining homeostasis and health during the whole lifespan. By contrast, an imbalance in the ecological composition of the gut microbiota, often termed “dysbiosis”, paves the way to several diseases, including metabolic disorders which involve glucose homeostasis.

With the worldwide increasing burden of diseases involving glucose homeostasis and insulin resistance (including pre-diabetes and type 2 diabetes—T2D), the knowledge about the relationships between gut microbiota-generated SCFAs and glucose homeostasis can offer new tools for clinical management and prevention. In this regard, in this review we assessed the latest evidence linking SCFAs with host metabolic health with special focus on the interplay between intestinal microbiota, dietary fiber, generation of SCFAs from fiber, and the effect of SCFAs on glucose homeostasis, as confirmed by clinical studies. 

## 2. Methodology

In this narrative review, we explored the scientific literature by PubMed analysis (PubMed (nih.gov)), scanning for international papers in English language (both observational and interventional studies) exploring the relationships between gut microbiota, SCFAs, and glucose metabolism, including pre-diabetes and type 2 diabetes (T2D). 

## 3. The Gut, the Microbiota, Fiber, and SCFAs

### 3.1. Microbiota and Intestinal Barrier

Gut microbiota are part of the complex and dynamic machinery known as the “intestinal barrier”, functionally generated by the interplay between different layers (Table 1).

Microbes, which include bacteria, viruses, fungi, bacteriophages, and protists, spread across the human intestine and form the microbial barrier. In health, microbes grow within the central mucin layer far from the surface of the enterocytes [6]. The human gut microbiota consists of trillions of these microbes which form a complex ecosystem that ranges between 10^3^–10^4^ per gram in the stomach, 10^5^–10^6^ in the jejunum, 10^8^–10^9^ in the terminal ileum, and about 10^12^–10^14^ bacteria per gram of gut content in the colon [7], containing at least 1000 different species of known bacteria, and carries 150 times more microbial genes than the human genome [8]. The second layer of the intestinal barrier is the mucin, consisting of heavily glycosylated proteins secreted by the intestinal goblet cells. A third “functional” layer is the combination of gastrointestinal motility involving the stomach, the gallbladder, and intestine, plus the secretion of gastric acid, hepatic bile, and pancreatic juice. A fourth layer is the epithelial barrier, consisting of the enterocytes, the Paneth cells, secreting antimicrobial peptides, and the goblet cells, secreting mucin.

Enterocytes are sealed by specific junctional proteins consisting of tight junctions found at the apical end of junctional complexes. Tight junctions are complex and dynamic structures, contacting neighboring intestinal epithelial cells comprising multiple protein families, such as occludins, claudins, and junctional adhesion molecules (JAMs). They bind intracellular membrane proteins, the zonula occludens (ZOs), which connects the transmembrane tight junction to the actin skeleton. 

Adherens junctions are found below the tight junction and are composed of transmembrane proteins, as follows: E-cadherin and nectin, linked to the cytoskeleton by scaffolding proteins, and catenin and afadin, connected to actin filaments. 

Desmosomes are transmembrane liner glycoproteins, desmoglein and democollin, which are cadherin proteins linked to intermediate keratin filaments [9]. 

In the context of the paracellular barrier, gap junctions are complex structures forming clusters of intercellular channels allowing direct diffusion of ions and small molecules between adjacent epithelial cells through multiple pathways, such as a pore pathway and a leak pathway. 

A fifth layer is the immune barrier, which consists of antimicrobial peptides, Toll-like receptors (TLRs), microbiota- and pathogen-associated molecular patterns (MAMPs and PAMPs, respectively), lymph nodes, and B/T lymphocytes. A sixth barrier is the gut vascular barrier consisting of endothelium associated with pericytes and enteric glial cells, tight junctions, and adherens junctions. The last barrier is the liver barrier, consisting of hepatic macrophagic Kupffer cells and stellate cells [6].

In this multilayer, inter-dependent, complex, and dynamic intestinal environment, the main functions of microbiota include the synthesis of micronutrients, the digestion and metabolism of nutrients, such as carbohydrates, fat, proteins, and vitamins, and the biotransformation of hepatic primary bile acids to secondary intestinal bile acids that play a key role in fat digestion and act as potent signaling hormone-like agonists. The host develops a natural immunity and tolerance towards the microbiota. This interaction influences the development and maturation of several cells within lymphoid tissues of the intestinal immune system [10,11]. In general, interventions with prebiotics, probiotics, and diet modify the intestinal microbiota. Notably, the bioavailability of substrates into the intestinal lumen drives the shape and the activity of gut microbes harboring or suppressing the presence of specific bacterial patterns at equilibrium with the resulting metabolites [12,13]. Fecal communities cluster into enterotypes, distinguished primarily by levels of Bacteroides and Prevotella. Enterotypes are strongly associated with long-term diets, particularly protein and animal fat (Bacteroides), versus carbohydrates (Prevotella) [14].

### 3.2. Fiber

Dietary nutrients are in close contact with the intestinal polymicrobial community and dietary fibers work as key substrates to generate SCFAs [15,16]. Fibers are carbohydrate polymers and oligomers encompassing monosaccharides, linked themselves to final molecules of different sizes. Other chemical groups or molecules, such as acetyl- and methyl-, can be linked to the carbon chains. Fibers are characterized by a large structure heterogeneity and various physical properties in terms of solubility or viscosity. The main food groups and fiber varieties according to solubility appear in Table 2. Fibers can be classified as dietary from whole food fibers and synthesized functional fibers [17]. Dietary fibers derive from grains, fruits, vegetables, and legumes [18]. The insoluble dietary fibers include cellulose, some hemicellulose, and lignin [19], which act mainly as mass-forming agents in gut transit [20]. Soluble fibers include wheat dextrin, pectin, gums, β-glucan, psyllium, fructans, and some hemicellulose [19].

### 3.3. Fiber as a Source of SCFAs in Health

The enzymatic profile of the human gut is not sufficient to completely metabolize dietary fiber. Soluble fiber, resistant to digestion by the host, represent the microbiota-accessible carbohydrates (MACs) [21]. The microbiota has a great gene-encoding power and is essential to guarantee the complete fiber digestion through the cecal and colonic fermentation [22]. This capacity of digestion depends on the richness in glycoside hydrolases (more than 260) of the huge microbial population, compared with only 17 enzymes in humans [23], which break down various types of carbohydrates [24]. This process involves more than 100 trillion bacteria. An MACs-rich diet in humans is associated with an increased content of colonic and fecal SCFAs [3,25,26,27,28,29,30], which are microbial metabolites and major products from bacterial anaerobic fermentation of dietary fiber and resistant starches in the intestine. Major SCFAs include acetic, propionic, butyric, valeric, and caproic acids, accounting for between two and six carbon (C) units. Among them, acetate (C2), propionate (C3), and butyrate (C4) are the most abundant SCFAs in the human body and the most abundant anions in the colon. Their synthesis involves different pathways, which include the Wood–Ljungdahl pathway for acetate, two molecules of acetate for butyrate, and the acrylate, succinate, and propanediol pathways for propionate (Figure 1) [31]. 

Acetate, propionate, and butyrate act as post-biotic molecules [32] and in the colon they exist in a molar ratio of 60:20:20, respectively; although, the relative proportion of each SCFAs depends on the substrate, the microbiota composition, and the gut transit time [33,34] (Figure 2 and Table 3). About 500–600 mM of SCFAs is produced in the large intestine daily, depending on the dietary fiber content. Psyllium and gums are not included into this process. 

The three SCFAs are produced in small amount (<1%) from amino acid metabolism [38]. Fecal excretion of SCFAs accounts for about 5% of total SCFAs. Microbial species are involved in shaping the SCFAs profile (Table 3). Bacteria species intake lactate and succinate and convert them into propionates [37]. Abundance of Bacteroides spp. is associated with production of propionate [39] and acetate [40], while butyrate is produced mainly by the Firmicutes phylum [40]. Colonic fermentation of fiber to SCFAs decreases pH levels, increases fecal acidification, and increases the growth and diversity of the gut microbiota taxa [41]. 

SCFAs act as mediators in several pathways, which include local, immune, endocrine effects, and microbiota–gut–brain communication.

As reported in Figure 3, following intestinal bacteria digestion of fermentable microbiota-accessible carbohydrates (MACs) consisting of fiber and starches, SCFAs are absorbed by colonocytes via passive diffusion or via active transport, mediated by H+-dependent monocarboxylate transporters (MCTs), previously known as solute carrier family (SLC) transporters. MCT1 (also known as SLC16a1) transports SCFAs in an H+-dependent, electroneutral manner. The SCFA anion transport occurs via the electrogenic, sodium-dependent monocarboxylate transporter 1 (SMCT1; also known as SLC5A8) [42]. 

At a cellular level, SCFAs promote complex and integrated effects, modulating the homeostasis and function of intestinal epithelial cell (Table 4).

In the enterocyte, acetate, butyrate, and propionate are converted to acetyl-CoA or propynyl-CoA via acetyl-CoA carboxylases (ACSSs) or β-oxidation to produce ATP via the citric acid cycle. This pathway contributes to maintain cell homeostasis including the function of the apical tight junctions [23]. SCFAs are also involved in cellular metabolism and immunity. In vitro, the three SCFAs promote intracellular permeability and the mechanism involved the modification of tight junction expression or distribution and zonulin (ZO-1) [44]. SCFAs potently stimulate ANGPTL4 (fasting-induced adipose factor—Fiaf) in human colon cell lines via PPARγ [45].

Butyrate provides the energy source to the gut mucosa. Propionate contributes to the liver gluconeogenesis and acetate achieves the highest systemic serum concentrations. Both propionate and butyrate provide the strongest inhibition of histone deacetylases [38]. 

Acetate in the gut epithelial cells of the animal model directly activates the nucleotide-binding oligomerization domain 3 (NLRP3) inflammasome, which increases the release of IL-18 [46], engaging the epithelial IL-18 receptor and promoting intestinal barrier integrity [47]. 

Butyrate provides energy, maintains the integrity of colonocytes, enhances the barrier function [48], and increases mucin expression, via MUC2 gene expression [49]. The protective effect on tight junction likely involves the activation of AMP-activated protein kinase (AMPK) [50] or the downregulation of claudin 2 expression [51]. This latter pathway involves the hypoxia-inducible factor-1, which modulates the efficiency of epithelial tight junction CLDN1, the gene coding for claudin-1 [52,53]. Butyrate induces Treg differentiation and plays a role in the control of inflammation [54]. Notably, administration of butyrate to ulcerative colitis patients decreases fecal calprotectin, a marker of gut inflammation [55]. 

The protective effects of propionate are also evident on the intestinal barrier. In the mice model fed with dextran sulfate sodium (DSS), propionate 1% counteracted the inflammatory changes due to DSS-induced colitis manifesting by weight loss, colonic damage, the increase in FITC-dextran in serum and the decrease in zonula occludens-1 (ZO-1), occludin, and E-cadherin expression in the colonic tissue. In addition, propionate inhibited the expression of interleukin (IL)-1β, IL-6, and tumor necrosis factor-alpha (TNF-alpha) mRNA and phosphorylation of signal transducer, the activator of transcription 3 (STAT3), reduced the myeloperoxidase level, and increased the superoxide dismutase and catalase level in the colon [56]. 

SCFAs protect the intestinal barrier by additional mechanisms which involve plasma membrane cholesterol-rich microdomain in T84 and Caco-2 cell cultures challenged with physiological concentrations of SCFAs [57]. In addition, SCFAs interact with the epithelial Toll-like receptors (TLRs) and activate the nuclear factor-κB signaling pathway which regulates the integrity of gut epithelial cells [46]. In colonic organoid from human colonic mucosal biopsies, the fermentable substrate 2′-O-fucosyllactose caused the increase in bifidobacteria and butyrate, with upregulation of claudin-5, a marker of enhanced barrier function [58]. Studies in humans showed that shift workers are prone to circadian oscillation and have decreased gut-derived plasma SCFAs, a finding which correlates with increased colonic permeability [59].

Several systemic effects of SCFAs originate from the intracellular effects of SCFAs which bind the G protein-coupled receptors (GPCRs), such as the free fatty acid receptor 2 (FFAR2, earlier known as GPR43) and FFAR3 (earlier known as GPR42), as well as GPR109A/HCAR2 (hydrocarboxylic acid receptor) and GPR164. These receptors are expressed in colonocytes, enterocytes, and in several other cells in the body, including neural cells. At the level of enteroendocrine L cells, the SCFAs-mediated stimulation of receptors is associated with the release of two anorexic hormones, glucagon-like peptide-1 (GLP-1) and the peptide YY (PYY). Notably, this pathway is independent of the bile acid pathway involving the ileal membrane receptor G protein-coupled bile acid receptor 1 (GPBAR-1) [6,11,26]. The SCFA-related pathways of hormone release are associated with several effects on appetite and food intake via systemic circulation or vagal afferents [60,61]. In addition, the incretin GLP-1 slows gastric emptying and gut transit, helps energy absorption [62], and enhances glucose-dependent insulin release [63]. Additional effects on gut motility likely involve the SCFAs-induced serotonin release from enterochromaffin cells [64]. 

Intracellular SCFAs have epigenetic effects, mainly through an inhibitory effect on histone deacetylases (HDAC) and an hyperacetylation of histones in lysine residues present in nucleosome [65]. This pathway is expressed in the gut, the associated immune tissue [66,67], and in the peripheral nervous system and the CNS [68,69]. 

SCFAs by acting on signaling pathways HDAC and GPR4 can influence the function of intestinal immune T cells 3 [70,71].

Although further studies are required, there is evidence suggesting additional roles for SCFAs in activation of brown adipose tissue and appetite control [72], body energy homeostasis [73], and even regulation of mitochondrial function [74].

In the colonocytes, SCFAs are partly oxidized to CO_2_, which can enter the portal blood. SCFAs which are not metabolized in the colonocytes enter the portal circulation at a concentration of about 260 μM (acetate), 30 μM (propionate), and 30 μM (butyrate). Once in the liver, SCFAs become substrates for energy production of hepatocytes via oxidation [75]. Acetate is also used for the synthesis of fatty acids and cholesterol [76]. Propionate can become a substrate for gluconeogenesis in the liver [77,78,79]. 

Only a minor fraction of the colon-derived acetate (36%), propionate (9%), and butyrate (2%) is transferred to the systemic circulation and peripheral tissues [80], with plasma concentrations of 25–250 μM, 1.4–13.4 μM, and 0.5–14.2 μM for acetate, propionate, and butyrate, respectively [81]. Acetate is also converted into acetyl-CoA in peripheral muscles for lipogenesis or oxidation. 

## 4. Manipulation of Intestinal Microbiota, SCFA and Effect on Glucose Homeostasis

The literature clearly points to a pivotal involvement of microbes in SCFAs’ metabolism [38,82]. A high microbiota diversity is associated with digestion of many types of complex carbohydrates with production of several SCFAs and additional gut microbiota diversity [26]. The interaction between gut microbiota and diet governs the qualitative and quantitative production of SCFAs. 

Potential tools to manipulate the intestinal microbiota include the ingestion of live beneficial bacteria as probiotics [83,84] or prebiotics as substrates selectively utilized by host microorganisms, conferring a health benefit [85]. The intestinal microbiota will ferment the prebiotic into SCFAs. In addition, diet will provide substrates able to manipulate the microbial composition. As mentioned earlier, plant-based foods, which are typical of the Mediterranean diet, will increase the availability of fermentable substrates by specific bacteria leading to SCFAs formation [86]. The intestinal microbiota is amenable to modulation by prebiotics [31,81,87,88,89], probiotics [66,90], or dietary regimens, which include the adherence to a Mediterranean diet [91,92,93]. Such approaches can substantially increase the colonic production of SCFAs. The ultimate clinical outcome of such approaches requires further studies.

In addition, following dietary changes, including habitual fiber intake, can select fiber-fermenting microbes, which, in turn, will inhibit other harmful species [94]. Production of SCFAs will influence different metabolic pathways [95] and therefore will contribute to prevent and modulate T2DM [2,96]. Further evidence is required in this field; therefore, herein, we discuss the main human studies addressing the relationship between the composition of the gut microbiota, concentration of SCFAs, and glucose metabolism. 

Among humans undergoing a controlled-feeding study, a high-fat/low-fiber or low-fat/high-fiber diet within 24 h were associated positively with Bacteroidetes and Actinobacteria in fat and a negative association between Firmicutes and Proteobacteria [14]. By contrast, an unbalanced microbiota, unable (or less capable) to metabolize dietary fiber, negatively impacts the hosts’ overall health. The reduced microbiota diversity decreased the availability of digestible carbohydrates, and certain SCFAs, such as propionate, can abnormally increase, and several functions, including metabolic effects, will decrease or become impaired. 

Results from animal models suggest that the composition of the gut microbiota can influence metabolic disorders, such as obesity, insulin resistance, and T2DM. In accordance with this hypothesis, prediabetic or T2DM patients often show intestinal dysbiosis, compared with healthy subjects [97,98]; although, specific microbial clusters are still lacking. The profile of gut microbiota plays an important role in T2DM [99] and steps include the maintenance of integrity of intestinal barrier. This step implies decreased chance for bacteria translocation and endotoxemia-dependent systemic inflammation, two changes likely involved during the early stage of disease. Patients with T2DM have reduced abundance of butyrate-producing bacteria [100]. SCFA supplementation in patients with T2DM increased the abundance of butyrate-producing bacteria and increased GLP-1 and hemoglobin A1c levels [101]. 

In line with the above-mentioned background, ongoing research is focusing on strategies able to modulate the human gut microbiota, since this approach represents a potential tool to prevent both metabolic and inflammatory diseases. Where genetic factors play a limited role in shaping the microbiota composition [102], environmental factors, which include the short-term effect of fecal transplantation [103] and the more stable effects of dietary habits [94], are potent tools in this respect. 

## 5. Clinical Studies on the Effects of Microbiota Profile and Dietary Manipulations in the Maintenance of Glucose Homeostasis and in T2DM 

A series of observational studies spanning from 2010 to 2020 show that the host metabolic state can profoundly influence the gut microbiota profile [86,91,92,100,104,105,106,107,108,109] (Table 5), affecting the microbiota/SCFA relationships, both in terms of maintenance of glucose homeostasis and pathological conditions. 

### 5.1. Effects on Glucose Homeostasis

In the large study carried out by Sanna et al. [107], 952 normoglycemic individuals underwent a genome-wide genotyping, gut metagenomic sequencing, and SCFA analyses. The authors adopted a bidirectional Mendelian randomization (MR) analysis to assess causality and found how the host’s genetically driven increase on butyrate was associated with improved insulin response after an oral glucose-tolerance test. By contrast, abnormalities in the production or absorption of propionate were associated with an increased risk of T2DM, suggesting a causal effect of the gut microbiome on metabolic traits.

A deeper investigation into the field of dietary intake contributes was reached evaluating the degree of adherence to the Mediterranean diet (MD), which usually involves a high-fiber intake. This additional focus was investigated in some studies enrolling healthy individuals with a low compared with high adherence to MD, or vegetarian individuals. These studies evaluated the gut microbiota profiling volunteers split based on the degree of adherence to MD. Studies reporting an higher adherence to MD found a high abundance of Bacteroidetes in light to the detection of *Prevotella* [86,91], *Bifidobacteria* [109], or other bacteria commonly metabolizing fiber (e.g., *Roseburia*, *Prevotella*, and *Lachnospira*) [92]. In the same studies, fecal butyrate or fecal propionate concentrations increased in individuals more compliant to MD and, therefore, reporting a higher daily intake of fiber [86,91,92,109]. Following a fiber-enriched diet, changes can be even more complex. As an example, Mitsou et al. [109] found that subjects more compliant to MD had lower *Escherichia coli* counts, a higher *Bifidobacteria*:*E. coli* ratio, increased levels and prevalence of *Candida albicans*, greater molar ratio of acetate, higher defecation frequency, and a more pronounced gastrointestinal symptomatology compared with those reporting low adherence. The molar ratio of another SCFA to valerate was lower in the case of high adherence to the MD compared with the other two tertiles. By contrast, fast food consumption was characterized by suppressed representation of lactobacilli and butyrate-producing bacteria.

The importance of dietary pattern in shaping the gut microbiota was the aim of the study carried out by De Filippo et al. [108], which compared the composition of gut microbiota in children from Africa and Italy. Notably, *Bacteroidetes* (mainly *Prevotella* and *Xylanibacter*) were increased in African children, while Firmicutes were reduced, in response to a plant-based diet, including minor cereals, such as millet and sorghum, legumes, and vegetables. The enrichment of such bacterial taxa suggested that the rural African diet seems to harbor the growth of fiber-hydrolyzing bacteria (xylans, cellulose). The SFCA profiling carried out onto fecal samples confirmed a higher amount of total SCFAs in African samples according to a significantly higher fecal concentration of acetic, propionic, butyric, and valeric acids. By contrast, Italian children who mainly consumed an animal-based diet, rich in fat and protein, had a decreased abundance of Bacteroidetes, while reporting more Firmicutes. Over this intriguing suggestion, it is important to consider the influence of the genetic background. With a specific focus onto the phylum Bacteroidetes, Italian children mainly accounted in *Bacteroides*, instead of *Prevotella*, and *Xylanibacter* as found for African children, further supporting what was previously found by Arumugam et al. [110].

Even short-term treatments might also provide beneficial effects with some fiber-enriched foods. Kovatcheva-Datchary et al. [111] and Nilsson et al. [112] compared the barley kernel-based bread, containing fiber 37.6 g/day, to wheat bread containing fiber 9.1 g/day in healthy volunteers for only 3 days. A brief consumption of barley kernel-based bread was associated with increased *Prevotella* and reduced *Bacteroides* abundance, confirming how these two genera are competitors for the same substrates. Additionally, even if not significantly, the genus *Prevotella* showed a greater relationship than *Bacteroides* with other genera recognized for metabolizing polysaccharides (i.e., *Dorea* and *Roseburia*) [111]. Moreover, values of correlation with the same taxa were higher than ones resulting from *Bacteroides*. Concerning the influence of the barley kernel on the host’s metabolism, in both studies, authors documented the reduction in postprandial glucose response [111,112], while only one of them showed increased total serum SCFAs concentration [112].

With the similar purpose to increase evidence about effects linked to short-term consumptions of fiber, David et al. [93] randomized 10 healthy individuals in a cross-over controlled 4-day trial. Subjects followed both a Western diet (enriched in animal foods, such as meat, eggs, and cheese, and a low fiber intake, equal to 9.36 ± 2.1 g/1000 kcal) or a plant-based diet, enriched in wholegrains, legumes, fruits, vegetables, and fiber, equal to 25.6 ± 1.1 g/1000 kcal. Compared with the Western diet, the plant-based diet resulted in increased plant-polysaccharide metabolizing bacteria belonging to *Lachnospiraceae* and *Ruminococcaceae* (such as *Roseburia*, *Ruminococcus*, and *Eubacterium*) at the same level of *Prevotella* among *Bacteroidetes*. In this study, despite the short term of evaluation, the variation of the above-mentioned taxa (which all increased) resulted in a higher concentration of fecal SCFAs butyrate and acetate. 

A similar outcome occurred in the study of Freeland et al. [111,113]. Randomizing hyperinsulinemic subjects (in accordance with fasting plasma insulin > 40 pmol/l) in two arms—a high- fiber wheat cereal diet against a low-fiber cereal diet—the profiles of metabolites were collected every 3 months for 12 months. No significant differences were found until 6 months into the study. Only at 9 months, instead, were acetate and butyrate concentrations higher for participants on the high-fiber than the control diet. Meanwhile, at the end of the trial (i.e., 12 months), authors detected an increase in GLP-1 secretion. Therefore, although without an immediate effect, this evidence suggested a contribution of fiber to a reduced risk for T2D.

### 5.2. Effects on Pre-Diabetes and T2DM

Wu et al. [105] found that the overall gut microbiota parallelly shifted according to the glycemic status in humans showing insulin resistance (not fasting glucose), and irrespective of diabetes treatment (metformin). In this study, according to KEGG orthologs (KO) annotations resulting from metagenomic analyses of fecal samples, bacterial genes involved in the butyrate metabolism were reduced in pre-diabetes and T2DM. 

Zhang et al. [104] and Wu et al. [105] designed two large studies with three subgroups of individuals consisting of subjects with normal glucose tolerance, subjects with pre-diabetes, and T2DM individuals. The body mass index increased in accordance with unhealthy status and the study analyzed the composition of fecal microbiota. Based on collected microbiota profiles, the authors speculated on a higher presence of butyrate-producing bacteria in normal glucose tolerance subjects than individuals with either pre-diabetes or T2DM. This conclusion was supported by the highest abundance of Roseburia (OTU1900), *Akkermansia muciniphila* ATCCBAA-835, and *Fecalibacterium prausnitzii* L2-6 in normal glucose tolerance subjects [104]. Nonetheless, *Lachnospiraceae*, *Ruminococcus*, *Eubacterium,* and overall *Clostridiales* were found at the highest abundance in the T2DM group and even these taxa are known to be within the main butyrate producers, as well as saccharolytic microorganisms [114]. Therefore, considering that no analyses were carried out to determine the SCFAs concentration, these findings were not sufficient to reach an absolute conclusion about those individuals, which were mainly colonized by butyrate-producing bacteria. Differently, a clear trend was observed for *Verrucomicrobia* that was reduced in prediabetic volunteers and even more in T2DM compared with NGT, to be claimed as a potential microbial marker of T2DM. 

Further studies based on metagenomic sequencing found increased *Roseburia* and *Fecalibacterium prausnitzii* in normal glucose tolerance individuals differently than patients with T2DM, in which the abundance of these taxa decreased [100,106]. Of note, similar findings were collected in spite of studies sampling feces from subjects with a different ethnicity (in Europe by Karlsson et al. [106] and in China by Quin et al. [100]). Both these studies [100,106] adopted the same statistical methodology to investigate fecal microbial genes. By combining the observations, we learn that metagenomic clusters of bacterial genes identify T2DM patients more accurately than the taxonomic assignment. The analysis of fecal microbiome allowed a better classification of women with impaired glucose tolerance, providing a potential tool to identify individuals at high risk of developing T2DM. Studies suggest the existence of changes in the intestinal environment of T2DM patients, which are likely to change in different geographical areas, even if maintaining a similar core. In T2DM subjects, this conclusion was reached thanks to similar findings in terms of gene contribution but given by different actors, as found for *Akkermansia muciniphila* by the first study [106], and for *lactobacilli* and *clostridial* species in the latter study [100].

Few interventional trials have tested the effect of dietary modulation of microbiota by using high-fiber diets or fiber-rich foods with the intent to ameliorate glucose metabolism [93,101,111,112,115,116,117,118] (Table 6). In general, a wide variability is typical of such studies, in terms of type of design (parallel, crossover), duration of study (from 3 days to 1 year), study groups (healthy individuals, subjects with metabolic syndrome—MetS—or T2DM), and type of intervention (plant-based diets, MD, or other high-fiber diets, etc.). 

Nevertheless, the emerging concept is that high-fiber diets and fiber-rich foods genuinely improve glucose metabolism through pathways involves gut microbiota and increased SCFAs metabolism.

Vitale et al. [115] enrolled overweight/obese individuals to evaluate the effects of an 8-week MD enriched in fiber (19.3 ± 3.1 g/1000 kcal). The regimen was associated with increased scores of alpha-diversities of fecal microbiota related to an increased abundance of different species, such as *Akkermansia muciniphila* and *Intestinimonas butyriciproducens*, and an increased concentration of plasma butyrate, postprandially. Notably, in individuals with high cardiometabolic risk, postprandial glucose and insulin sensitivity also improved when compared with the control diet, consisting of fiber 8.1 ± 2.3 g/1000 kcal. Butyrate concentrations directly correlate with postprandial insulin sensitivity.

Zhao et al. [101] treated T2DM patients with a 12-week high-fiber diet enriched in fiber 37.1 ± 1.9 g. The control diet consisted of half amount of fiber. Compared with baseline, the high-fiber diet caused increased levels of fecal butyrate concentrations associated with a reduced concentration of fasting glucose and HbA1c. The whole genome shotgun of fecal microbiota showed a greater gene-encoding power to metabolize fiber mainly determined by increased abundances of *Fecalibacterium prausnitzii* and *Bifidobacterium pseudocatenulatum*. Notably, the fecal microbiota transplant into gnotobiotic recipients (C57BL/6J mice) confirmed the microbiota-related and metabolomic findings obtained from the in vivo evaluation.

Haro et al. [116] studied 20 obese male subjects over 1 year. In a crossover design, authors used a low-fat/high-fiber diet consisting of 14.1 ± 0.2 g/1000 kcal (mainly from wholegrains) and an MD enriched in fiber (12.9 ± 0.2 g/1000 kcal, mainly from vegetables and nuts). No common findings were found in terms of microbiota variations that underlined a strict relationship occurring from specific taxa and the type of diet. Additionally, only few differences were found in plasma metabolites, as well as fecal ones. Focusing on the abundance of the main recognized SCFA-producing bacteria, the genus *Roseburia* decreased in the low-fat/high-fiber diet, while the MD enhanced *Roseburia* and *Oscillospira* genera. A deeper evaluation (at species level) revealed an increase in *Fecalibacterium prausnitzii* in the low-fat/high-fiber diet only. Therefore, although collected results emphasized a specific relationship between microbes and these two different diets without compromising the hosts’ metabolomic profiles, both regimens improved insulin sensitivity as a result of the significant consumption of fiber.

In a crossover study, Hald et al. [117] investigated, in MetS subjects, the effects of arabinoxylan (from whole grain rye and wheat bran) and resistant starch type 2 (from raw potato starch and maize starch) on intestinal microbiota and also included a specific focus onto SCFAs. Whole grains are of interest because of their high fiber content and fermentative effect, as in the case of cereal fiber arabinoxylans and bran. Collected outcomes were compared against a low-fiber Western-style diet. Volunteers fed the fiber-enriched diet reported a significant decrease in indices of alpha diversity related to fecal microbiota community. This resulted from a decrease in various taxa (e.g., *Bacteroides*, *Odoribacter*, *Dorea*, *Lachnospira*, *Ruminococcus*, and *Eubacterium*), while only Bifidobacterium has been harbored by the fiber-enriched diet. Despite the few differences in microbial taxonomy, the study revealed how the fiber-enriched diet determined a greater fecal concentration of total SCFAs (mainly acetate and butyrate) than the low-fiber Western-style diet. Oppositely, the latter diet determined a greater concentration of brain chain fatty acids (BCFAs), indicating a high intestinal fermentation of proteins (mainly iso-butyrate and iso-valerate). 

Vetrani et al. [118] studied 40 MetS individuals during a 12-week whole grain-based diet containing cereal fiber 28.9 ± 1.1 g/day compared with a refined cereal-based diet used as control (cereal fiber: 11.8 ± 0.4 g/day). Starting from no differences in SCFAs concentration at baseline between the two arms, the whole grain-diet was associated with an increased plasma propionate concentration correlating with improved insulin postprandial response.

## 6. Mechanisms of Action of Dietary Manipulation

Fiber in diet can influence glucose metabolism as shown in both healthy individuals and T2DM individuals. Fiber viscosity, water solubility, and fermentation rates represent important properties in this respect [119]. In addition, dietary fiber modulates microbial composition and some metabolites, including SCFAs with ultimate effects on glucose metabolism (Figure 4).

SCFAs act as secretagogues for two key intestinal hormones, namely glucagon-like peptide-1 (GLP-1) and peptide YY (PYY). This step appears to increase the satiety feeling through the gut–brain axis. By this pathway, SCFAs indirectly reduce appetite and food intake, a step which prevents body weight gain and, in turn, the risk of T2DM. The SCFAs effect on GLP-1-mediated increase in insulin secretion can regulate blood glucose concentrations [33].

SCFAs can decrease hepatic glycolysis and gluconeogenesis and increase glycogen synthesis. SCFAs also increase long chain fatty acids oxidation [33,121,122,123,124].

In skeletal muscle and adipose tissue SCFAs improve glucose uptake, an effect mediated by increased expression of GLUT4, through AMP kinase (AMPK) activity. In the skeletal muscle, SCFAs reduce glycolysis, a step associated with accumulation of glucose-6-phosphate and increased glycogen synthesis [121,122,123,124,125,126,127].

These studies are not conclusive and deserve confirmation; however, further evidence suggests that daily supplementation with 10 g inulin-propionate is associated with increased GLP-1 and PYY and reduced food intake [128,129]. In addition, daily supplementation with 4 g sodium butyrate improves insulin sensitivity; however, the effect is evident in lean subjects and not in subjects with metabolic syndrome [130].

The fine interplay between host and microbiota includes intestinal aspects involving production of microbial metabolites and preservation of intestinal integrity. If mechanisms fail, dysbiosis, leaky gut, and endotoxemia become pathological aspects. Evidence suggests that lactic acid bacteria (LAB), as well as other taxa, such as *Akkermansia muciniphila* and *Bifidobacteria*, can reduce intestinal permeability and inflammation [95,131]. In this scenario, SCFAs are bacterial post-biotics made from fermentation of dietary fiber. In line with this evidence, a diet enriched in fiber, following a prebiotic effect, drives the gut microbiota mainly harboring SCFA-producing microbial groups (e.g., LAB [31]) and taxa, such as *Roseburia, Blautia, Fecalibacterium prausnitzii*, and *Prevotella* [132,133,134]. Moreover, evidence suggests that the gut microbiota is involved in the regulation of glucose metabolism, and fiber intake is one protective factor. These mechanisms can therefore play a role with respect to the risk of developing T2DM [135].

Fiber intake might become an additional “natural” tool used for the prevention and management of metabolic disorders. Studies need to check if soluble and insoluble fiber may affect microbiota in a different way. Fiber in diet include β-glucan and arabinoxylans from wholegrains, pectins from fruit, vegetables, and legumes, and resistant starch. Therefore, the gut microbiota could distinctively process fiber and influence glucose homeostasis. In this respect, soluble, readily fermented fiber should be more effective than other types of fiber [119]. The effect of insoluble fiber on T2DM risk might differ and involve additional mechanisms, such as control of body weight gain, and increased fecal excretion of glucose [119,136]. Thus, the consumption of wholegrain, legumes, fruit, and vegetables should be increased in patients with pre-diabetes or diabetes, since this approach will increase dietary fiber intake. 

By contrast, there is no evidence for fiber supplements or SCFA-based formulations.

Whether special—rather than traditional—dietary regimens provide similar or different effects on the fiber–microbiota–SCFA axis is still under debate. For example, short-term carbohydrate-restricted diets are associated with a reduction in SCFA-producing microbial species (*Bifidobacteria*) and SCFAs [137,138]. Ketogenic diets might also dramatically influence the gut microbiota profile [139]. By literature review, Rondanelli et al. [140] suggests that the very low calorie ketogenic diet preserved the core fecal microbiome but altered the composition of fecal microbial populations in relation to the plasma metabolome and fecal bile acid composition. The weight loss resulted in a reduction in *E. rectale* and *Roseburia* and an increase in *Christensenellaceae* and *Akkermansia*. Not all studies found a decrease in *Fecalibacterium prausnitzii*; however, in this field of research, it is important to emphasize the great impact of the adopted sequencing methods [141]. 

## 7. Conclusions

An increasing number of studies are actively investigating how diet and other external stimuli drive the gut microbiota composition and activity. Quality and quantity of dietary fiber play a pivotal role and act as microbiota-accessible carbohydrates and substrates for production of SCFAs, namely acetate, butyrate, and propionate. SCFAs, in turn, produce effects both at the local intestinal level and at a systemic level, acting through epigenetic mechanisms and via interaction with several receptors and tissues involved in the maintenance of glucose homeostasis, in pre-diabetes and in T2DM. 

Specific links have been identified between microbiota diversity, SCFAs production, and glucose homeostasis, with possible implications in terms of management and prevention of altered glucose metabolism and T2DM. In this context, several clinical studies show the beneficial effect of fiber-enriched diets on health and in patients with pre-diabetes or T2DM, mainly in terms of increased insulin sensitivity, reduced fasting, and postprandial glucose. 

The exclusion from the diet of whole grains, such as pasta and bakery products, may not be beneficial for healthy microbiota patterns. Thus, an increased daily fiber intake might be a valid tool to improve microbiota composition and activity, and to prevent metabolic disorders. In addition, the role of prebiotics, probiotics, and SCFAs, per se, in improving abnormalities in the maintenance of glucose homeostasis is a matter of active research.

## Figures and Tables

**Figure 1 ijms-23-01105-f001:**
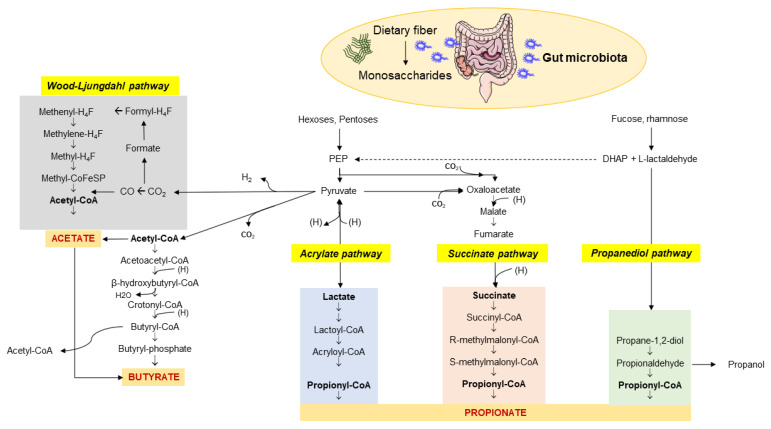
Pathways involved in the biosynthesis of SCFAs from dietary fiber and carbohydrate fermentation by the colonic microbiota. The three major SCFAs are: (1) acetate which originates via the Wood–Ljungdahl pathway or acetyl-CoA; (2) butyrate synthesized from two molecules of acetyl-CoA; (3) propionate from PEP involving the acrylate pathway or the succinate pathway or the propanediol pathway after microbial transformation of fucose and rhamnose. Abbreviations: PEP—phosphoenolpyruvate; DHAP—dihydroxyacetone phosphate. Adapted from Kho et al. [31].

**Figure 2 ijms-23-01105-f002:**
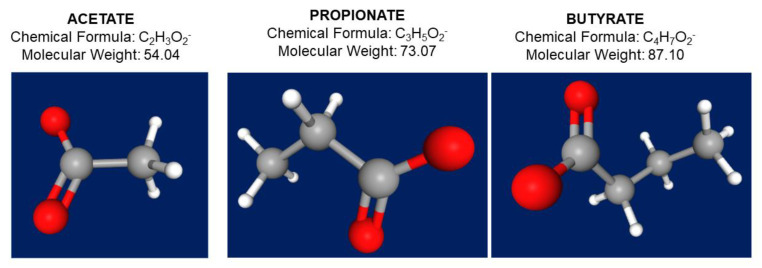
Chemical formula, molecular weight, and 3D structure of the three main short chain fatty acids acetate (C2), propionate (C3), and butyrate (C4). In the 3D structure, atoms appear as hydrogen in white color, carbon as grey color, and oxygen as red color. https://pubchem.ncbi.nlm.nih.gov/ (accessed 18 January 2022).

**Figure 3 ijms-23-01105-f003:**
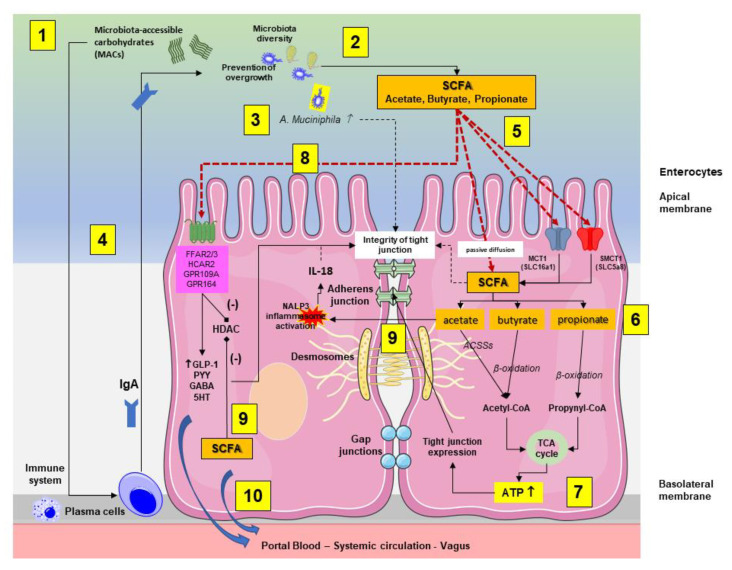
Interplay between the microbiota-accessible carbohydrates (MACs), the gut microbiota, the production short chain fatty acids (SCFAs), and the enterocytes (mainly colonocytes). The main pathways involved are summarized in the two enterocytes. (**1**) Following initial digestion and intestinal transit, the dietary MACs meet the gut microbiota, which is characterized by high and physiological diversity and no bacterial overgrowth. (**2**) SCFA-producing bacteria, mainly in the colon, will digest MACs and increase the luminal content of SCFAs. (**3**) In this environment, the abundance of *A. muciniphila* increases and is associated with protective effects on mucin and tight junction integrity. (**4**) In addition, a diet enriched in MACs will positively stimulate the immune system, leading to plasma cell-mediated production of immunoglobulins A (IgA) with further control on microbiota function, diversity, and prevention of overgrowth. (**5**) In the colonocyte, SCFAs are absorbed by colonocytes via passive diffusion or via active transport mediated by H+-dependent monocarboxylate transporters (MCTs). (**6**) The SCFAs acetate, butyrate, and propionate are converted to acetyl-CoA or propynyl-CoA by pathways involving the acetyl-CoA carboxylases (ACSSs) and beta oxidation. (**7**) This step produces ATP, which contributes to the maintenance of cell homeostasis, including the function of tight junctions. (**8**) Via stimulation of receptors at the apical membrane, SCFAs promote the secretion of gut hormones, such as glucagon-like peptide 1 (GLP1) and peptide YY (PYY), γ-aminobutyric acid (GABA), and serotonin (5-HT). At this level, butyrate inhibits (-) histone deacetylases (HDACs) with consequent anti-inflammatory effect by reducing NF-κB-induced pro-inflammatory mediators, such as TNF-α, IL-6, IL-12, and iNOS [43]. (**9**) Intracellular SCFAs contribute to inhibition (-) of HDAC. Acetate activates the inflammasome nucleotide-binding oligomerization domain 3 (NLRP3) with secretion of the protective IL-18 from epithelial cells, which maintains the tight junction’s function. (**10**) Colon-derived SCFAs reach the systemic circulation promoting anti-inflammatory and immunomodulatory effects as well as increasing insulin secretion, maintaining energy homeostasis, and improving liver and brain function.

**Figure 4 ijms-23-01105-f004:**
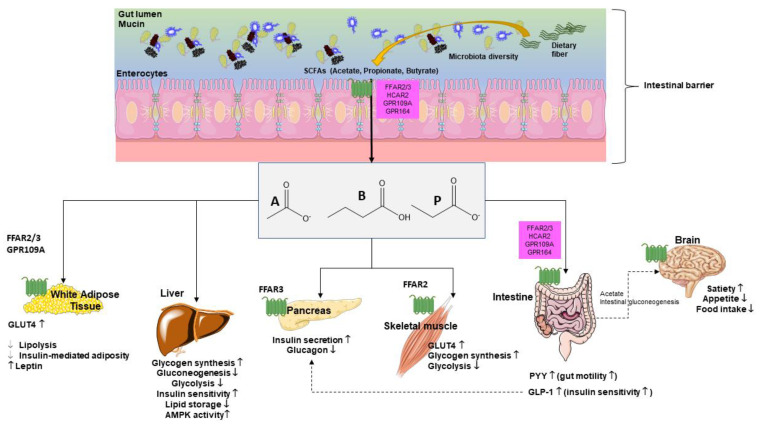
Glucose metabolism is influenced by short chain fatty acids (SCFAs), systemically, at various levels [120]. Main target organs include adipocytes, liver, pancreas, skeletal muscle, intestine, and brain where pathways govern mechanisms, which include receptors, synthesis, hormones, and perception. Abbreviations: GLP-1—glucagon-like peptide-1; GLUT-4—activated glucose transporter protein-4; GPR—G-protein-coupled receptors; PYY—peptide YY; A—acetate; P—propionate; B—butyrate.

**Table 1 ijms-23-01105-t001:** Different physical and anatomical layers contributing to the intestinal barrier.

Microbial barrier (gut microbiota). Gut mucus, accumulating at the interface between the intestinal lumen and the brush border of enterocytes. Functional barrier, which is interplay between gastrointestinal motility and gastric acid, and biliary and pancreatic secretions. Epithelial barrier and tight junctions (enterocytes). Immunological barrier which is the combination of the immune-competent cells and their products. Gut–vascular interface. Liver barrier which represents the hepatic filter.

Adapted from Portincasa et al. [5].

**Table 2 ijms-23-01105-t002:** Main food groups and fiber varieties according to solubility.

Food Group	Soluble Fibers	Insoluble Fibers
Cereals and grains	Nonstarch polysaccharides	Nonstarch polysaccharides
	Hemicellulose Arabinoxylan β-glucan	Hemicellulose Cellulose Lignin
	Resistant oligosaccharides	Resistant starch
	Inulin	
Fruits and vegetables	Nonstarch polysaccharides	Nonstarch polysaccharides
	Hemicellulose Pectin	Hemicellulose Cellulose Pectin
	Resistant oligosaccharides	Lignin
	Inulin	Resistant starch
Legumes and pulses	Nonstarch polysaccharides	Nonstarch polysaccharides
	Hemicellulose Pectin Gum	Hemicellulose Pectin
		Lignin
		Resistant starch

Adapted from Swann et al., 2019 [18], and the Institute of Medicine, 2005 [17].

**Table 3 ijms-23-01105-t003:** Dietary source, fiber substrates, and SCFA-producing bacteria.

Dietary Source	Substrates	Fermenting Genera
Cashew, green banana, white beans, oat, and potato	Resistant starch	*Ruminococcus*, *Bacteroides*
Seaweed and cereal bran	Cellulose	*Bacteroides*, *Ruminococcus*
Cereal bran	Hemi-celluloses (xylan and arabinoxylan)	*Bacteroides*, *Roseburia*, *Prevotella*
Apples, apricots, cherries, oranges, and carrots	Pectin	*Eubacterium*, *Bacteroides, Fecalibacterium*
Asparagus, leek, onions, banana, wheat, garlic, chicory, and artichoke	Fructans (inulin and fructooligosaccharides)	*Bacteroides*, *Fecalibacterium*
Breast milk	Milk oligosaccharides	*Bifidobacterium*
Milk, yogurt, buttermilk, and cheese	Lactose (only in lactose-intolerant people)	*Bifidobacterium*
Oat, barley, wheat, rye, mushrooms, and seaweed	β-Glucan	*Eubacterium*, *Atopobium, Enterococcus*, *Lactobacillus, Prevotella*, *Clostridium* cluster XIVa
Acacia tree and prepared food additive	Gum arabic	*Bifidobacterium*, *Lactobacillus*, *Ruminococcus*
Guar bean and prepared food additive	Guar gum	*Bifidobacterium*, *Ruminococcus*
Seaweed	Laminarin	*Prevotella*
Artichoke, beans, beetroot, broccoli, chickpeas, fennel, lentils, lettuce, radicchio, and onion	Galacto-oligosaccharides	*Bifidobacterium*
Cottonseed flour, soy flour, onions, chickpeas, beans, peas, and lentils	Raffinose and stachyose	*Bifidobacterium*, *Lactobacillus*

Adapted from [35,36,37].

**Table 4 ijms-23-01105-t004:** Effects of SCFAs on intestinal epithelial cell homeostasis and function.

Energy substrate for ATP production. Receptor activation, mainly G protein-coupled receptors. Maintenance of barrier function and integrity. In particular, modulation of apical tight junctions, activation of NLRP3 inflammasome, increased mucin expression, anti-inflammatory effects, interaction with epithelial Toll-like receptors, and activation of the nuclear factor-κB signaling pathway. Modulation of immunity and control of inflammation. Treg differentiation, modulation of inflammation mediators. Modulation of intracellular permeability. Epigenetic effects, with inhibition of histone deacetylases, hyperacetylation of histones, and modulation of gene expression.

**Table 5 ijms-23-01105-t005:** Observational studies pointing to the role of SCFAs in the maintenance of glucose homeostasis, in pre-diabetes and T2DM.

Author, Year	Study Groups	Number (M/F)	BMI (Kg/m^2^)	Major Findings
Wu et al., 2020 [105] 2020	Healthy individuals	206 (106/100)	28.2	In pre-diabetes and patients with T2DM ↓ abundance of butyrate-producing bacteria. Findings are independent of fasting glucose or treatment with metformin treatment.
	Pre-diabetes	220 (107/113)	28.3	
	T2DM	58 (32/27)	31.6	
Sanna et al., 2019 [107]	Healthy individuals selected on the basis of SCFAs and genome, fecal metagenomic sequences.	952	-	Butyrate-producing bacteria have a protective role against T2DM.
Garcia-Mantrana et al., 2018 [86]	Healthy individuals	27 (11/16)	19–28	Evaluation by food frequency questionnaire. High adherence to Mediterranean diet associated with ↑ abundance of *Bifidobacteria,* ↑ *Bacteroidetes*. ↓ abundance of *Firmicutes*:*Bacteroidates* ratio. ↑ Fecal SCFAs and propionate. High animal protein intake associated with ↓ abundance of *Bacteroidetes*, ↑ *Firmicutes*:*Bacteroidates* ratio.
Mitsou et al., 2017 [109]	Healthy individuals	116 (61/55)	25–30	Evaluation by food frequency questionnaire. High adherence to Mediterranean diet associated with ↑ fecal SCFAs.
Gutierrez-Diaz et al., 2016 [91]	Healthy individuals with low adherence to Mediterranean diet.	10 (2/8)	21.2–31.2	Evaluation by food frequency questionnaire. High adherence to Mediterranean diet associated with ↑ abundance of fecal SCFAs, ↑ faecal propionate and butyrate.
	Healthy individuals with high adherence to Mediterranean diet.	21 (8/13)	21.6–31.0	
De Filippis et al., 2016 [92]	Vegetarian individuals	51	19.4–24.4	Vegetarian diet, vegan diet, and omnivore; high adherence to Mediterranean diet associated with ↑ abundance of *Prevotella* and ↑ fecal propionate.
	Vegan individuals	51	19.1–23.5	
	Omnivore individuals	51	20.1–24.1	
Zhang et al., 2013 [104]	Healthy individuals	44	23.4	In healthy individuals ↑ abundance of butyrate-producing bacteria (*Akkermansia muciniphila* ATCCBAA-835, and *Fecalibacterium prausnitzii* L2-6). In patients with pre-T2DM ↓ abundance of *Bacteroides* and *Verrucomicroniae.*
	Pre-diabetes	64	24.9	
	T2DM	13	26.5	
Karlsson et al., 2013 [106]	Healthy individuals	43 (0/43)	20–40	In healthy individuals ↑ abundance of *Roseburia* ↑ *Fecalibacterium prausnitzii.* In patients with T2DM ↓ decreased *Roseburia* ↓ *Fecalibacterium prausnitzii.*
	Pre-diabetes	49 (0/49)		
	T2DM	53 (0/53)		
Quin et al., 2012 [100]	Healthy individuals	182	18–40	In healthy individuals ↑ abundance of *Roseburia* and ↑ *Fecalibacterium prausnitzii.* In patients with T2DM ↓ abundance of *Roseburia* and ↓ *Fecalibacterium* prausnitzii in individuals with T2DM.
	T2DM	183		
De Filippo et al., 2010 [108]	African children (1–6 yrs)	14 (9/5)		Evaluation by food frequency questionnaire. Plant-based diet associated with ↑ abundance of *Prevotella* and ↑ *Xylanibacter* ↓ *Firmicutes.* Animal-based diet associated with ↓ abundance of *Prevotella* ↓ *Xylanibacter* ↑ *Firmicutes.*
	Italian children (1–6 yrs)	15 (9/6)		

Abbreviations: ↓—significant decrease; ↑—significant increase; BMI—body mass index; SCFAs— short chain fatty acids; T2DM— type 2 diabetes.

**Table 6 ijms-23-01105-t006:** Randomized clinical trials pointing to the role of SCFAs in the maintenance of glucose homeostasis, in metabolic syndrome, and in type 2 diabetes.

Author Year	Study Groups	Number (M/F)	BMI Kg/m^2^	Study Design	Duration	Intervention	Major Findings
Vitale et al., 2021 [115]	At least one criterion of MetS (overweight/obesity)	29 (14/15)	25–35	Parallel	8 weeks	Mediterranean diet consisting of fiber 19.3 g/1000 kcal compared with control diet consisting of fiber 8.1 g/1000 kcal)	↑ *Intestinimonas butyriciproducens* ↑ *Akkermansia muciniphila* ↑ Plasma butyric acid ↓ Postprandial glucose ↓ Postprandial insulin ↑ Oral glucose insulin sensitivity
Zhao et al., 2018 [101]	Type 2 DM	43	25–35	Parallel	12 weeks	High-fiber diet consisting of fiber 37.1 g compared with control diet consisting of fiber 16.1 g	↑ Fecal butyrate ↓ HbA1c ↓ Fasting glucose
Haro et al., 2016 [116]	MetS (insulin sensitivity/obesity)	20 (20/0)	30–40	Parallel	1 year	Mediterranean diet consisting of fiber: 12.9 ± 0.2 g/Kcal mainly from vegetables compared with high-fiber diet consisting of fiber: 14.1 ± 0.2 g/1000 kcal, mainly form wholegrains	↑ *Roseburia* ↓ *Prevotella* ↑ Insulin sensitivity index High-fib diet ↓ *Roseburia* *↑ Prevotella*
Hald et al., 2016 [117]	MetS	19	25.9–41	Crossover	4 weeks	Diet enriched with Arabinoxylan and Resistant starch consisting of fiber 64 g compared with Western diet consisting of fiber 17.6 g	↑ *Bifidobacteria* ↑ Fecal SCFAs ↑ Fecal butyrate
Vetrani et al., 2016 [118]	MetS (overweight/obesity and type 2 diabetes)	40 (16/24)	25–35	Parallel	12 weeks	Wholegrain diet consisting of total fiber 40 g with fiber from cereal 28.9 g, compared with refined cereal diet consisting of total fiber 22.1 g with fiber from cereal 11.8 g	↑ Plasma propionate ↓ Postprandial insulin
Kovatcheva-Datchary et al., 2015 [111]	Healthy individuals	39 (6/33)	18–28	Parallel	3 days	Barley kernel-based bread consisting of fiber 37.6 g compared with white wheat bread consisting of fiber 9.1 g	↑ *Prevotella:Bacteroides* ↓ Postprandial glucose
Nilsson et al., 2015 [112]	Healthy individuals	20 (3/17)	18–28	Crossover	3 days	Barley kernel-based bread consisting of fiber 37.6 g compared with white wheat bread consisting of fiber 9.1 g	↑ Plasma SCFAs ↓ Glucose ↓ Insulin
David et al., 2014 [93]	Healthy individuals	10 (5/5)	19–32	Crossover	4 days	Plant-based diet consisting of fiber 26 g/1000 kcal compared with Western diet consisting of fiber 9.3 g/1000 kcal	↑ *Prevotella* ↑ *Roseburia* ↑ Fecal butyrate Western diet ↓ *Prevotella* ↑ *Bacteroides*
Freeland et al., 2010 [114]	Hyperinsulinaemic individuals	28	24–27	Parallel	1 year	High-wheat fiber cereal consisting of 24 g fiber/day compared with low-fiber cereal	↑ Acetate and butyrate concentrations ↑ Plasma GLP-1

Abbreviations: ↓—significant decrease; ↑—significant increase; BMI—body mass index; HbA1c—glycated hemoglobin; GLP-1—glucagon-like peptide-1; MetS—metabolic syndrome; SCFAs—short chain fatty acids.

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
