# Peer review of "Gut Microbiota and Short Chain Fatty Acids: Implications in Glucose Homeostasis"

_ijms, 2022, doi:10.3390/ijms23031105_

Round 1

Reviewer 1 Report

I would like to thank the authors for sharing their efforts. I suggest employing the PRISMA checklist to improve the reproducibility of your work, and to attach it to the revised manuscript. 

Author Response

We thank the reviewer for his/her comments. Although the paper is a narrative review and not a systematic review, please find enclosed the PRISMA checklist.

Reviewer 2 Report

The manuscript entitled “Gut Microbiota, Short Chain Fatty Acids and Implications in Glucose Homeostasis” is a comprehensive review on the role of microbiota-SCFA in health and diseases. The authors discuss the interplay between intestinal microbiota, dietary fiber and SCFA in maintaining homeostasis and whether they could be modified to improve certain pathology. Further, the manuscript references 137 publications highlighting current knowledge on the topic.

Several suggestions listed below will significantly improve the manuscript, and benefit the field and readers.

  1. The manuscript would significantly benefit from additional editing.
  2. The writing style is overly simplified and misleading. For example: “Tight junctions are composed of three transmembrane proteins”. However, tight junctions are complex and dynamic strictures, contacting neighboring intestinal epithelial cells comprised of multiple protein families such as occludins, claudins, JAMs…
  3. Similarly, the statement “Gap junctions develop as tunnel for small molecules passing between adjacent epithelial cells” is misleading. The paracellular barrier, to the flux of ions and molecules, is complex structure that include multiple pathways such as a pore pathway and a leak pathway.
  4. There are statements throughout the manuscript that lower the scientific depth. For example, in the conclusion, the statement: “the consumption of pasta and bakery products, in form of whole grains, according to an increased daily fiber intake work to harbor healthy dietary microbial pat-terns and for this reason it might be a valid tool to improve microbiota composition”. Please modify the statement with a scientific narrative such as “exclusion certain food may not be beneficial (such as whole grains) for healthy microbiota patterns….”
  5. Well described barrier layers if presented in a table, would benefit the manuscript and readers.
  6. It is unclear why Figure 2 shows two chemical structures. It is repetitive and one would be sufficient.
  7. Model in Figure 3 is unclear. Improve the model and explain what mechanisms these two intestinal epithelial cells present.
  8. A table providing what effect SCFA has on intestinal epithelial cell function (cell receptors, energy function, barrier function, immune function, epigenetics, transcription) will be beneficial.
  9. Sections 5 and 6 substantially discuss current findings. However, the contribution of microbiota-SCFA in homeostatic vs pathological connections is less clear.
  10. The abbreviation for type 2 diabetes, e.g. T2DM is used for the first time in line 297, but defined later in line 314. Please define abbreviations when used for the first time in the manuscript.

Author Response

Reviewer 2

The manuscript entitled “Gut Microbiota, Short Chain Fatty Acids and Implications in Glucose Homeostasis” is a comprehensive review on the role of microbiota-SCFA in health and diseases. The authors discuss the interplay between intestinal microbiota, dietary fiber and SCFA in maintaining homeostasis and whether they could be modified to improve certain pathology. Further, the manuscript references 137 publications highlighting current knowledge on the topic.

Several suggestions listed below will significantly improve the manuscript, and benefit the field and readers.

  1. The manuscript would significantly benefit from additional editing.

The manuscript has been now extensively revised

  1. The writing style is overly simplified and misleading. For example: “Tight junctions are composed of three transmembrane proteins”. However, tight junctions are complex and dynamic strictures, contacting neighboring intestinal epithelial cells comprised of multiple protein families such as occludins, claudins, JAMs…

The writing style has been now revised throughout the manuscript. The quoted sentence has been rewritten as suggested and changes appear in red (Line 97-101).

  1. Similarly, the statement “Gap junctions develop as tunnel for small molecules passing between adjacent epithelial cells” is misleading. The paracellular barrier, to the flux of ions and molecules, is complex structure that include multiple pathways such as a pore pathway and a leak pathway.

The sentence has been reformulated as follows (Line 107-110):

“In the context of the paracellular barrier, gap junctions are complex structures forming clusters of intercellular channels allowing direct diffusion of ions and small molecules between adjacent epithelial cells through multiple pathways such as a pore pathway and a leak pathway”

  1. There are statements throughout the manuscript that lower the scientific depth. For example, in the conclusion, the statement: “the consumption of pasta and bakery products, in form of whole grains, according to an increased daily fiber intake work to harbor healthy dietary microbial pat-terns and for this reason it might be a valid tool to improve microbiota composition”. Please modify the statement with a scientific narrative such as “exclusion certain food may not be beneficial (such as whole grains) for healthy microbiota patterns….”

The quoted sentence has been now rewritten as follows:

“The exclusion from diet of whole grains (as pasta and bakery products) may not be beneficial for healthy microbiota patterns. Thus, an increased daily fiber intake might be a valid tool to improve microbiota composition and activity, and to prevent metabolic disorders.”

  1. Well described barrier layers if presented in a table, would benefit the manuscript and readers.

We thank the reviewer for this comment. A new Table 1 has been now added in the manuscript. Please consider that a new reference has been added, recently published from our group. Thus, all other references have been renumbered.

Table 1. Different physical and anatomical layers contributing to the intestinal barrier

  • Microbial barrier (gut microbiota)
  • Gut mucus, accumulating at the interface between the intestinal lumen and the brush border of enterocytes
  • Functional barrier, which is interplay between gastrointestinal motility and gastric acid, biliary and pancreatic secretions
  • Epithelial barrier and tight junctions (enterocytes)
  • Immunological barrier which is the combination of the immune-competent cells and their products
  • Gut-vascular interface
  • Liver barrier which represents the hepatic filter

Adapted from Portincasa et al. [5]

  1. It is unclear why Figure 2 shows two chemical structures. It is repetitive and one would be sufficient.

Figure 2 has been now revised including only one chemical structure.

  1. Model in Figure 3 is unclear. Improve the model and explain what mechanisms these two intestinal epithelial cells present.

Thank you very much for mentioning this point. Figure 3 has been now revised and mainly involved mechanisms have been listed as 10 different steps, fully explained in the legend. The figure and the leged, we agree, are much clearer now.

  1. A table providing what effect SCFA has on intestinal epithelial cell function (cell receptors, energy function, barrier function, immune function, epigenetics, transcription) will be beneficial.

We thank the reviewer for this comment. A new Table 4 has been now added in the manuscript:

Table 4. Effects of SCFA on intestinal epithelial cell homeostasis and function

  • Energy substrate for ATP production
  • Receptor activation, mainly G protein-coupled receptors
  • Maintenance of barrier function and integrity. In particular modulation of apical tight junctions, activation of NLRP3 inflammasome, increased mucin expression, anti-inflammatory effects, interaction with epithelial Toll-like receptors and activation of the nuclear factor-κB signaling pathway
  • Modulation of immunity and control of inflammation. Treg differentiation, modulation of inflammation mediators
  • Modulation of intracellular permeability
  • Epigenetic effects with inhibition of histone deacetylases, hyperacetylation of histones, modulation of gene expression

  1. Sections 5 and 6 substantially discuss current findings. However, the contribution of microbiota-SCFA in homeostatic vs pathological connections is less clear.

We thank the reviewer for this comment. In the revised version of the manuscript, two new sub-paragraphs have been added, and the text has been remodulated accordingly:

5.1 Effects on glucose homeostasis

5.2 Effects on prediabetes and T2DM

  1. The abbreviation for type 2 diabetes, e.g. T2DM is used for the first time in line 297, but defined later in line 314. Please define abbreviations when used for the first time in the manuscript.

All the abbreviations have been now defined when used for the first time in the manuscript.

Reviewer 3 Report

The article by Portincasa et al. aims to  explain the association between gut microbiota, short-chain fatty acids and glucose homeostasis

Major comments:

The title talks about glucose homeostasis, however, there is a bit of a mess in this regard. The authors speaks about  metabolic diseases in tables, in the title about glucose homeostasis and in text about type 2 diabetes (T2D).

Metabolic disorder, include others than T2D, for example, obesity or metabolic syndrome). Specify which ones it deals with or if only T2DM is concerning this work.

This should be defined as the content of the review becoming imprecise and very confusing. The authors should reorganize the content of this review and decide which metabolic diseases to talk about. If they are only going to focus on T2D, delimit and correct the text accordingly.

In this regard the conclusion section should be more thematically precise and provide a little more discussion.

The introduction seems more like a summary than an introduction to the paper. It should be clear what this review contributes what others do not and the importance of the chosen topic.

 Why have the authors decided to write this paper?

Minor comments:

 The methodology described in this way is more suitable for systematic reviews.

Figure 2: the figure caption should be corrected. The name given is not correct.

The first is the formula is structural and the second is the empirical formula.

Tables 3 and 4:

Tables 3 and 4 should be joined in one. In addition, it should be specified which metabolic diseases they are (Metabolic disorder, include others, for example, obesity or metabolic syndrome). Specify which ones it deals with or if only T2D in included.

The author and year should be in the same column.

Indicate if the study is conducted in women, men or both.

CONCLUSIONS: Only the first letter should be capitalized.

Fiber and fibre are used insistingly. Please correct it.

Line 60-62: Reference 4 should be updated.

 Define SCFA  the first time it is used

Figure 3: The figure caption is too long. The text should be included in the main body of the manuscript.

Line 314 TD2, this abbreviation should be defined the first time it is used. In general, review all abbreviations in the text and when they are defined

Author Response

Reviewer 3

We thank this reviewer for her/his constructive comments which helped us to improve this manuscript. Here we provide the point-by-point reply to her/his comments.

The article by Portincasa et al. aims to explain the association between gut microbiota, short-chain fatty acids and glucose homeostasis

Major comments:

The title talks about glucose homeostasis, however, there is a bit of a mess in this regard. The authors speaks about metabolic diseases in tables, in the title about glucose homeostasis and in text about type 2 diabetes (T2D).

We thank the reviewer for this helpful comment. The text and tables have been now revised focusing on glucose homeostasis, prediabetes, and type 2 diabetes. The text is much clearer now.

Metabolic disorder, include others than T2D, for example, obesity or metabolic syndrome). Specify which ones it deals with or if only T2DM is concerning this work.

As mentioned in the previous answer, the paper now is targeting glucose homeostasis, prediabetes and T2DM. Other metabolic disorders have not been considered, and the text/tables have been modified accordingly. Again, thank you for mentioning this since the manuscript is much clearer now.

This should be defined as the content of the review becoming imprecise and very confusing. The authors should reorganize the content of this review and decide which metabolic diseases to talk about. If they are only going to focus on T2D, delimit and correct the text accordingly.

We agree with the reviewer. The text/sections have been now re-arranged focusing on the maintenance of glucose homeostasis, on prediabetes and T2DM.

In particular:

- Section 5 have been re-titled as follows: “5. Clinical studies on the effects of microbiota profile and dietary manipulations in the maintenance of glucose homeostasis and in T2D “.

- the text of section 5 has been re-written considering two new sub-sections 5.1 and 5.2:

5.1 Effects on glucose homeostasis

5.2 Effects on prediabetes and T2DM

- the title of Table 5 has been re-phrased as follows:

Table 5. Observational studies pointing to the role of SCFA in the maintenance of glucose homeostasis, in prediabetes and T2DM.

- the title of Table 6 has been re-phrased as follows:

Table 6. Randomized clinical trials pointing to the role of SCFA in the maintenance of glucose homeostasis, in metabolic syndrome and in T2DM.

- the conclusion section has been revised

In this regard the conclusion section should be more thematically precise and provide a little more discussion.

The conclusions section has been rewritten and expanded, focusing on pathogenetic and therapeutics links between gut microbiota, SCFA, and altered glucose homeostasis.

The introduction seems more like a summary than an introduction to the paper. It should be clear what this review contributes what others do not and the importance of the chosen topic.

We thank the reviewer for the critical opinion. As suggested, we modified the introduction section and we think that now is much improved.

 Why have the authors decided to write this paper?

We thank the reviewer for this suggestion. We added the following text in the last paragraph of the introduction section:

“With the worldwide increasing burden of diseases involving glucose homeostasis and insulin resistance (including prediabetes and type 2 diabetes, T2D), the knowledge about the relationships between gut microbiota-generated SCFA and glucose homeostasis can offer new tools for clinical management and prevention. In this regard, we assessed in this review the latest evidence linking SCFA with host metabolic health with special focus on the interplay between intestinal microbiota, dietary fiber, generation of SCFA from fiber, and the effect of SCFA on glucose homeostasis, as confirmed by clinical studies”

Minor comments:

 The methodology described in this way is more suitable for systematic reviews.

The methodology section has been now re-written as follows:

“We reviewed the scientific literature by Pubmed analysis (PubMed (nih.gov)) scanning for international papers in English language (both observational and interventional studies) exploring the relationships between gut microbiota, SCFA and glucose metabolism, including prediabetes and type 2 diabetes.”

Figure 2: the figure caption should be corrected. The name given is not correct. The first is the formula is structural and the second is the empirical formula.

Figure 2 has been now re-drawn. The caption has been re-written accordingly

Tables 3 and 4:

Tables 3 and 4 should be joined in one. In addition, it should be specified which metabolic diseases they are (Metabolic disorder, include others, for example, obesity or metabolic syndrome). Specify which ones it deals with or if only T2D in included.

The author and year should be in the same column.

Indicate if the study is conducted in women, men or both.

According to suggestions, we added the author’s name and year of publication in the same column. We also provided (when available) the number of enrolled Males/Females (as M/F). In addition, we specified (where applicable) the metabolic condition mentioned in each study. We would like to underline that reported findings are related to T2D directly (Insulin resistance, impaired fasting glucose, hyperinsulinemia etc..) or indirectly (overweight, obesity, hyperlipidemia ecc..).

We agree with the reviewer that the two tables (now Tables 5 and 6) might be joined. However, due to the complexity of both tables and the different topics explored (i.e., observational studies in Table 5, randomized clinical trials in table 6), we believe that it might be useful to the reader to maintain separate the two tables, and to distinctly discuss the two aspects in the text. We hope the reviewer agrees with this choice, for the sake of clarity.

CONCLUSIONS: Only the first letter should be capitalized.

Reviewer is right, we revised it.

Fiber and fibre are used insistingly. Please correct it.

The reviewer is totally right, we corrected “fibre” to “fiber” throughout the manuscript

Line 60-62: Reference 4 should be updated.

According to the reviewer suggestion, we updated the sentence and we added two more recent references

 Define SCFA  the first time it is used

The SCFA is defined as “Short chain fatty acid” at the first use (introduction section).

Figure 3: The figure caption is too long. The text should be included in the main body of the manuscript.

The reviewer is totally right, we shortened the caption with a brief description of which is already included in the main text

Line 314 TD2, this abbreviation should be defined the first time it is used. In general, review all abbreviations in the text and when they are defined

All the abbreviations have been now defined when used for the first time in the manuscript.

Round 2

Reviewer 1 Report

I would like to thank the authors for their reply. I still suggest specifying the selected format (i.e. narrative review) in the title, abstract and the "methodology" section. This way, the reader will instantly identify the review type and better evaluate the overall significance/replicability of its results. 

Author Response

Reviewer 1

I would like to thank the authors for their reply. I still suggest specifying the selected format (i.e. narrative review) in the title, abstract and the "methodology" section. This way, the reader will instantly identify the review type and better evaluate the overall significance/replicability of its results. 

We thank the reviewer for the positive reply. Changes appear now in underscored red color.

Many thanks for this comment. The format “narrative review” has been now indicated:

In the title:

“Narrative Review

Gut Microbiota, Short Chain Fatty Acids and Implications in Glucose Homeostasis”

Abstract (Line 26): “In this narrative review we discuss the relevant research focusing on the relationship between gut microbiota, SCFA, and glucose metabolism….”

Methodology section (line 63): “In this narrative review we explored the scientific literature by…”

Reviewer 3 Report

The manuscript has improved in this version.
Most of my issues have been addressed.
I have only a few minor comments for the authors
Figure 2: The molecular weight and formula for butyrate are not correct. The authors have represented butyric acid.
Propionate and acetate : The authors have represented  the propionate and acetate anions. Indicate it  or replace the negative charge by an R.
Heading of table 6: do not abbreviate T2DM or define the abbreviation at the bottom.
Line 644: type 2 diabetes mellitus, use abbreviation. 

Author Response

Reviewer 3

The manuscript has improved in this version.
Most of my issues have been addressed.
I have only a few minor comments for the authors

We thank the reviewer for the positive reply. Changes appear now in underscored red color.

Figure 2: The molecular weight and formula for butyrate are not correct. The authors have represented butyric acid.

Propionate and acetate : The authors have represented  the propionate and acetate anions. Indicate it or replace the negative charge by an R.

We apologize. We adapted the figure and its legend with the appropriate names: acetate, propionate, and butyrate. The legend is now the following, also reporting the web link of pubchem:

Figure 2. Chemical formula, molecular weight and 3D structure of the three main short chain fatty acids acetate (C2), propionate (C3) and butyrate (C4). In the 3D structure, atoms appear as hydrogen in white color, carbon as grey color, and oxygen as red color.https://pubchem.ncbi.nlm.nih.gov/

Heading of table 6: do not abbreviate T2DM or define the abbreviation at the bottom.
the abbreviation “T2DM” has been removed and replaced with “type 2 diabetes”

Line 644: type 2 diabetes mellitus, use abbreviation. 

“type 2 diabetes mellitus” has been now replaced with “T2DM”

In addition, we re-inserted Figure 4 which was unfocused in some points and not properly centered in the page.
